# Analyzing the Performance of Transformers for the Prediction of the Blood Glucose Level Considering Imputation and Smoothing

**Edgar Acuna** [1,*] ![ID], **Roxana Aparicio** [2] ![ID] **and Velcy Palomino** [3]

1 Mathematical Sciences Department, University of Puerto Rico at Mayaguez, Mayaguez PR00681, Puerto Rico
2 Computer Science Department, University of Puerto Rico at Bayamon, Bayamon PR00959, Puerto Rico
3 Computing and Information Sciences and Engineering, University of Puerto Rico at Mayaguez, Mayaguez PR00681, Puerto Rico
* Correspondence: edgar.acuna@upr.edu

**Abstract:** In this paper we investigate the effect of two preprocessing techniques, data imputation and smoothing, in the prediction of blood glucose level in type 1 diabetes patients, using a novel deep learning model called Transformer. We train three models: XGBoost, a one-dimensional convolutional neural network (1D-CNN), and the Transformer model to predict future blood glucose levels for a 30-min horizon using a 60-min time series history in the OhioT1DM dataset. We also compare four methods of handling missing time series data during the model training: hourly mean, linear interpolation, cubic interpolation, and spline interpolation; and two smoothing techniques: Kalman smoothing and smoothing splines. Our experiments show that the Transformer performs better than XGBoost and 1D-CNN when only continuous glucose monitoring (CGM) is used as a predictor, and that it is very competitive against XGBoost when CGM and carbohydrate intake from the meal are used to predict blood glucose level. Overall, our results are more accurate than those appearing in the literature.

**Keywords:** diabetes; Transformer; 1D-CNN; XGBoosting; glucose prediction; imputation; Kalman smoothing; smoothing splines

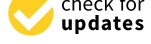



## 1. Introduction

Type 1 diabetes is a chronic disease in which the pancreas fails to produce insulin to regulate blood glucose (BG) levels [1,2]. This disorder can lead to both hypoglycemia (BG concentration < 70 mg/dL) and hyperglycemia (BG concentration > 180 mg/dL) and requires patients to self-regulate carbohydrate consumption and need to take insulin supplements. Furthermore, hyperglycemia increases the risk of heart disease and stroke and can lead to medical complications such as blindness, kidney failure, and amputations. Meanwhile, hypoglycemia can cause acute symptoms such as loss of consciousness, seizures, and even death [3]. To avoid such diabetic complications, patients continually monitor their BG levels and adjust insulin doses accordingly. An increasing number of type 1 diabetes patients are adopting continuous glucose monitoring (CGM) devices and insulin pump therapy, where a wearable device releases insulin subcutaneously to mimic pancreatic response. The insulin dose needed to regulate the BG level has to be controlled manually. Effective prediction of BG levels is necessary to give patients time to intervene and prevent complications. The ultimate goal is to improve the quality of life of diabetes patients.

In type 2 diabetes, the metabolic system can generate insulin, but the immune system creates resistance to it, resulting in a similar outcome as in type 1 diabetes. Subjects with type 2 diabetes do not always need to supply themselves with daily insulin doses. Usually, it is enough if they have a healthy routine and diet. This type of diabetes appears over a lifetime and can be prevented or delayed with a healthy lifestyle.

Blood glucose levels have a complex dynamic that depends on many different variables, such as carbohydrate intake, recent insulin injections, physical activity, stress levels, the presence of an infection in the body, sleeping patterns, hormonal patterns, etc. (Bremer and Gough [4]; Cryer et al. [5]). This complexity makes predicting short-term blood glucose changes a challenging task, and developing machine learning (ML) becomes an obvious approach to improving patient care.

Low-cost sensors continuously measuring blood glucose levels in intervals of a few minutes, combined with machine learning solutions, enable personalized precision health and diabetes management. In this study, we present a novel deep learning algorithm called Transformer [6] for predicting blood glucose levels up to one half-hour into the future for diabetic patients. The model outputs the prediction along with an estimate of its certainty, helping users to interpret the predicted levels. Furthermore, we compare the performance of Transformers with two other algorithms, XGBoosting [7] from the Machine Learning ecosystem, and 1D-CNN [8], a particular version of the CNN specifically for sequential data. Additionally, in our study, we evaluate the effect on the performance of the prediction algorithms of two pre-processing tasks: imputation and smoothing.

The paper is organized as follows. In Section 2, we present a comprehensive summary of previous research on blood glucose prediction. In Section 3 we describe in detail the two data preprocessing techniques and the three prediction algorithms used in our study. In Section 4, the results of our experiments are presented along with a discussion of them. Finally, Section 5 summarizes the conclusions and findings of this work. Additionally, future work regarding this study is mentioned.

## 2. Related Work

The dataset used in this paper is known as the OhioT1DM Dataset (see Marling and Burnescu [9]). It contains approximately eight weeks of data for 12 subjects (7 male and 5 female) with type 1 diabetes. The experiments were performed in two cohorts of six subjects each.

Martinsson et al. [10] applied a long short-term memory model (LSTM) using only CGM as a predictor to the six subjects of the first cohort from the OhioT1DM Dataset. The average RMSE obtained for the model was 20.1 +/− 2.5. They used only 30 min of history with a prediction horizon of 30 min, but they did not handle missing values.

Since a decade ago, people have been considering other factors, such as insulin and meals, to predict glucose levels. Zecchin et al. [11] used data consisting of CGM monitoring for three consecutive real-life days of 15 diabetes type 1 patients in an open-loop setup. CGM was measured by the Dexcom Seven Plus CGM sensor (Dexcom Inc, San Diego, CA, USA), which has a sampling period of 5 min. The patients manually recorded information on the dose of insulin injections and the carbohydrate content of meals. The authors concluded that meal carbohydrate information improves prediction significantly compared to insulin information. In Zecchin et al. [12] a jump neural network is used to predict glucose considering other factors. They used the mean absolute error (MAE) as a metric of accuracy instead of the root mean squared error (RMSE). The authors used data containing information from only 3 days.

Li et al. [3] proposed a deep learning algorithm for glucose prediction using a multi-layer convolutional recurrent neural network (CRNN) architecture. Clinical data used by the authors were obtained from a clinical study at Imperial College Healthcare NHS Trust St. Mary's Hospital, London (UK), consisting of multiple phases evaluating the benefits of an advanced insulin bolus calculator for T1D subjects. The dataset in consideration was collected from a 6-month period involving ten adult subjects with T1D. The information included in the dataset comprises glucose, meal, insulin, and associated time stamps. In building the dataset, the authors mainly consider CGM and self-reported data such as insulin boluses and meals. Before that, they excluded participants whose data exhibited significant gaps (corresponding to weeks of missing data), insufficient reports of exercise over the 6-month period, and extensive errors in sensor readings. The model is primar-

ily trained on CGM, carbohydrate, and insulin data. After preprocessing (filtering and alignment), the time-aligned multi-dimensional time series data of BG, carbohydrate, and insulin values are fed to CRNN for training. The architecture of the CRNN is composed of three parts: three convolutional layers that extract the data features using convolution and pooling, followed by a recurrent neural network (RNN) layer with LSTM cells and two fully connected layers [13]. The RMSE for the prediction model in the 10-subject real data collected during six months was 21.07 +/− 2.35.

Zhu et al. [14] applied a dilated CNN model to the first cohort from the OhioT1DM Dataset. They used CGM, meal, and bolus as glucose level predictors. Additionally, they applied several data preprocessing techniques. They obtained an RMSE 21.726 +/− 2.523. Chen et al. [15] also use three predictors and a dilated RNN model. After preprocessing the data, they obtained an average RMSE of 19.042 in the first cohort.

Midroni et al. [7] concluded that XGBoosting performs better than LSTM and Random Forest in predicting glucose level. Only the first cohort of the OhioT1DM dataset was used. Additionally, they conclude that many of the provided features do not improve glucose prediction. Furthermore, at least for XGBoosting, the only feature that matters for glucose prediction is CGM.

Rabby et al. [16] used CGM along with other factors: bolus, meal, and steps to predict glucose level. They proposed a stacked LSTM model followed by two layers of dense CNN after performing initially Kalman smoothing on the CGM data from the first cohort. They achieved an average RMSE of 6.45 mg/dL for 30 min of prediction horizon.

Deng et al. [17] used time series generative networks (TimeGAN). They applied their model using the twelve training datasets of both cohorts of the OhioT1DM and the six testing datasets of the first cohort as the training dataset. Additionally, each of the second cohort's six testing datasets was considered a test set. They only used CGM as the predictor feature. An average RMSE of 19.08 on the testing datasets was obtained. In this study, a non-personalized blood glucose level prediction was considered. This approach is risky as the glucose level of the patients in the OhioT1DM dataset does not seem to have the same behavior.

Bevan and Coenen [18] applied an LSTM with a 24-h attention mechanism model using training and test datasets as described above. Furthermore, they considered two approaches to handling missing data: discarding any training sequences with one or more missing datapoints and replacing missing values with zeros following standardization. They showed that the second approach may help the system learn to be robust to missing data. An average RMSE of 18.23 on the testing datasets was obtained.

Joedicke et al. [19] used a genetic programming model for both cohorts of the OhioT1DM dataset under the same setup as above. They compared their model with several other models among the ARIMA, Random Forest, and multivariate Linear Regression to predict future values of glucose after 30 min, as a function of basal value (bv), bolus dose (bd), basis GSR value (gsr), basis skin temperature (sk), bolus type (bt), and CGM. An average RMSE of 20.13 was obtained using the algorithm on the testing datasets.

## 3. Materials and Methods

### 3.1. Data Acquisition

The dataset used in this paper was provided by researchers from Ohio University (see Marling and Burnescu [9]), and it is known as the OhioT1DM Dataset. It contains approximately eight weeks of data for each of the 12 subjects (7 male and 5 female) having type 1 diabetes. All subjects were on insulin pump therapy with continuous glucose monitoring (CGM).

They wore Medtronic 530 G or 630 G insulin pumps and used Medtronic Enlite CGM sensors throughout the 8-week data collection period. The sensors reported life-event data via a custom smartphone app and provided physiological data from a fitness band. The data from the first six weeks constitute the training set and the remaining two weeks constitute the test set.

The first cohort of six individuals (2 male and 4 female) wore Basis Peak fitness bands. Data for this cohort were released in 2018. The second cohort of six individuals (5 male and 1 female) wore the Empatica Embrace. Data for this cohort are included in the 2020 release. For each subject, there is a training dataset as well as a testing dataset. Therefore, the whole dataset contains 24 data files, all in an xml format. Each data file contains information about the features in a column-wise manner.

From the xml files, the values of each feature need to be extracted before the analysis. Using a Python script, the data for each subject are extracted in several csv files, one for each feature. The first feature is the glucose level measured by the sensor every five minutes (CGM), along with the time when it was recorded. There are many missing values in this feature (see Table 1). In some subjects, there are even more than one-day gaps in the CGM's recording. In the data files below, the CGM's records show glucose levels measured manually and several other features. In this study, we have considered only one additional feature: meals. Unfortunately, some important features were not measured in the second cohort.

**Table 1.** Percentage of missing values in the CGM feature for each subject. In parentheses is the number of gap intervals exceeding six hours.

| Subject | Percentage of Missing Values in Training Dataset | Largest Gap in Training Dataset | Percentage of Missing Values in Testing Dataset | Largest Gap in Testing Dataset |
|---|---|---|---|---|
| 559 | 10.62 | 13 h 03 m (4) | 12.61 | 09 h 01 m (2) |
| 563 | 7.43 | 24 h 06 m (4) | 4.53 | 09 h 52 m (1) |
| 570 | 5.41 | 12 h 07 m (1) | 4.68 | 03 h 15 m (0) |
| 575 | 9.44 | 11 h 24 m (7) | 4.74 | 04 h 19 m (0) |
| 588 | 3.55 | 12 h 37 m (2) | 3.12 | 04 h 01 m (0) |
| 591 | 14.96 | 80 h 38 m (5) | 3.05 | 03 h 51 m (0) |
| 540 | 8.86 | 19 h 40 m (5) | 5.54 | 09 h 38 m (1) |
| 544 | 16.17 | 61 h 35 m (5) | 13.42 | 28 h 54 m (1) |
| 552 | 18.17 | 43 h 27 m (6) | 40.15 | 117 h 59 m (2) |
| 567 | 19.77 | 26 h 22 m (9) | 16.78 | 16 h 02 m (2) |
| 584 | 8.28 | 14 h 47 m (4) | 11.04 | 16 h 56 m (1) |
| 596 | 20.2 | 49 h 25 m (9) | 8.65 | 10 h 44 m (1) |

It is well-known that handling many missing values, say above 20 percent, can lead to bias in the prediction task. Based on data in Table 1, subject 552 was excluded from our study because they have a high percentage of missing values in the testing set (see Table 1). On the other hand, subject 596 was excluded because they have a high rate of missing values on the training set. Subject 544 was excluded because they have a high percentage of missing values in the training and testing datasets. For the same reasons, subject 567 was excluded from our study. Furthermore, for this last subject, the probability distribution of its test data seems to be somehow different from the training set. For instance, on Tuesdays there are few collected data, and the glucose levels are deficient in the test set.

Therefore, in our work, we will use only eight subjects from the OhioT1DM dataset: all six from the first cohort and two from the second cohort. From Table 1, subjects 588 and 570 have the lowest missing values among all the subjects of both cohorts.

*3.2. Data Analysis*

In this section, we explore the data by computing some statistical measures and visualizing the training sets of the selected subjects to be used in this paper. We aim to see if significant differences exist between the subjects chosen for our study. In Table 2, we compare the percentage of time when the subjects are in either hypoglycemic or hyperglycemic status.

**Table 2.** Time in range of diabetes patients under study.

| Subject | Training | | Testing | |
|---|---|---|---|---|
| | Hypoglycemic | Hyperglycemic | Hypoglycemic | Hyperglycemic |
| 559 | 4.16% | 39.23% | 3.00% | 37.11% |
| 563 | 2.57% | 23.29% | 0.7% | 38.83% |
| 570 | 1.97% | 54.92% | 0.4% | 69.63% |
| 575 | 8.76% | 22.28% | 5.37% | 31.12% |
| 588 | 1.05% | 35.20% | 0.14% | 45.39% |
| 591 | 3.94% | 32.00% | 5.18% | 27.54% |
| 540 | 7.07% | 20.43% | 4.97% | 32.52% |
| 584 | 0.93% | 50.36% | 1.02% | 36.67% |

From Table 2, subject 575 is the most hypoglycemic, whereas subject 588 is the least hypoglycemic. On the other hand, subject 570 is the most hyperglycemic, whereas subject 540 is the least hyperglycemic.

In Figure 1, boxplots of the CGM by day of the week for subject 584's training dataset show a cyclic pattern during the week with a peak on two days: Wednesday and Saturday. This pattern shows the seasonality of the CGM time series according to the day of the week.

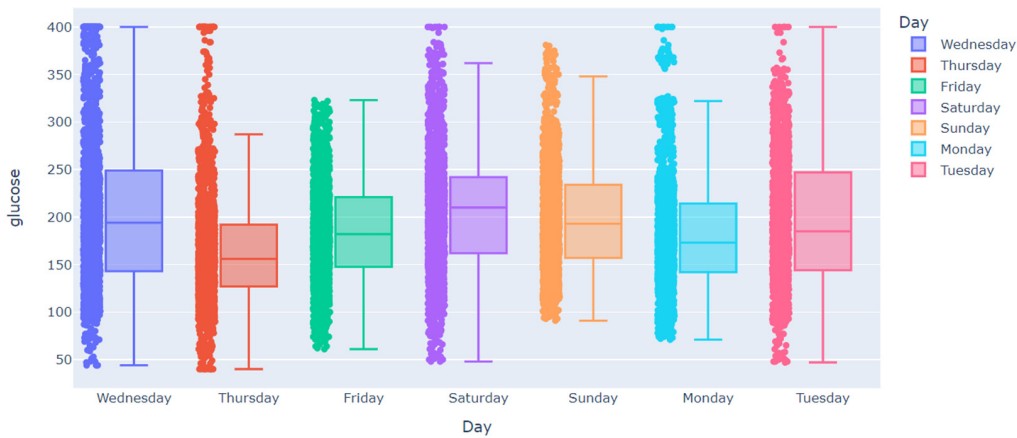

**Figure 1.** Boxplots of CGM by day of the week for the training data of the subject 584.

In Figure 2, boxplots of the CGM by day of the week for subject 575's training dataset show no cyclic pattern.

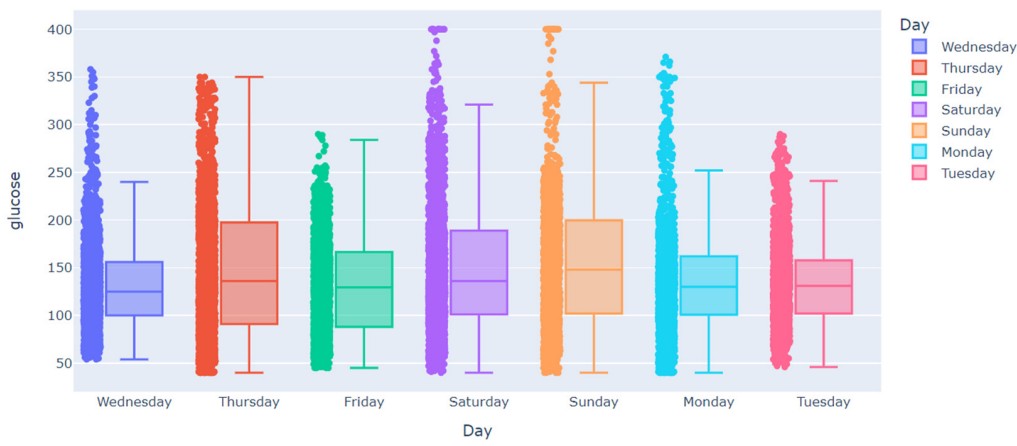

**Figure 2.** Boxplots of CGM by day of the week for the training data of the subject 575.

In Figure 3, boxplots of the CGM by hour of the day for subject 570′s training dataset show a cyclic pattern. This implies non-stationarity of the time series. Dataset 588 has a similar behavior, whereas datasets 584 and 591 also show some signs of non-stationarity.

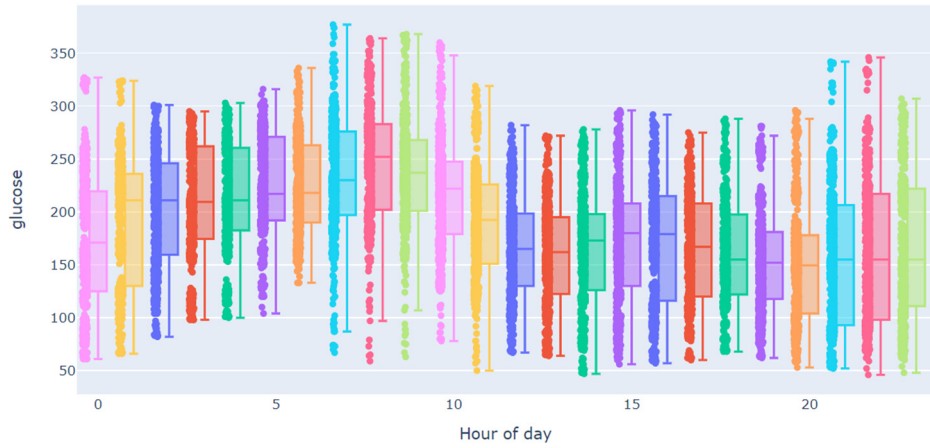

**Figure 3.** Boxplots of CGM by hour of the day for the training data of the subject 570.

In Figure 4, boxplots of the CGM by the hour of the day for subject 540′s training dataset do not show a cyclic pattern. This pattern indicates stationarity of the time series. Subjects 559, 563, and 575 have similar behavior.

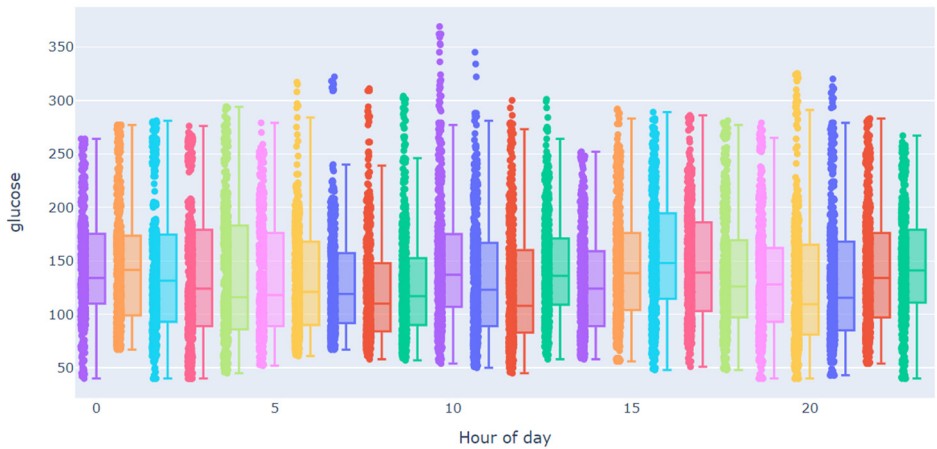

**Figure 4.** Boxplots of CGM by hour of the day for the training data of the subject 540.

### 3.3. Handling the Gaps

It was expected that the sensor would report the CGM data every 5 min; however, due to several reasons, the sequential data are not equally spaced. Therefore, the first step is to create equally spaced data, including NA values, when the data were not reported. After that, several actions can be taken before applying any model. In the following sections, we will describe these actions.

### 3.3.1. Removing Missing Values

In order to perform prediction with a time series, it is necessary to convert the time series data into supervised data. We used a Python script to perform such a task. Besides the equally spaced data, we consider the lookback time steps and the horizon timesteps as inputs for the script. By default, the script removes instances with missing data.

The missing values in CGM time series are imputed using four methods: hourly mean, linear interpolation, spline interpolation, and polynomial interpolation. Furthermore, Kalman smoothing, which involves imputing missing values and smoothing the time series, is also

considered. Finally, to compete with this last method, we used spline smoothing, but first we imputed the missing values with the hourly mean since spline smoothing can be applied only over complete data. Each of these methods are explained in detail the next section.

### 3.3.2. Imputation Methods

There are plenty of imputation methods for time series, but we have chosen only four that have given good results in previous studies [20].

(i)    Hourly mean.

A missing value of CGM at a given timestamp is replaced by the mean of the whole CGM time series in the hour that includes the timestamp corresponding to the missing value. Figure 5 shows the observed and imputed values by hourly mean for subject 588 in the week starting on 31 August and ending on 7 September. It appears that the imputation preserves the trend of the time series.

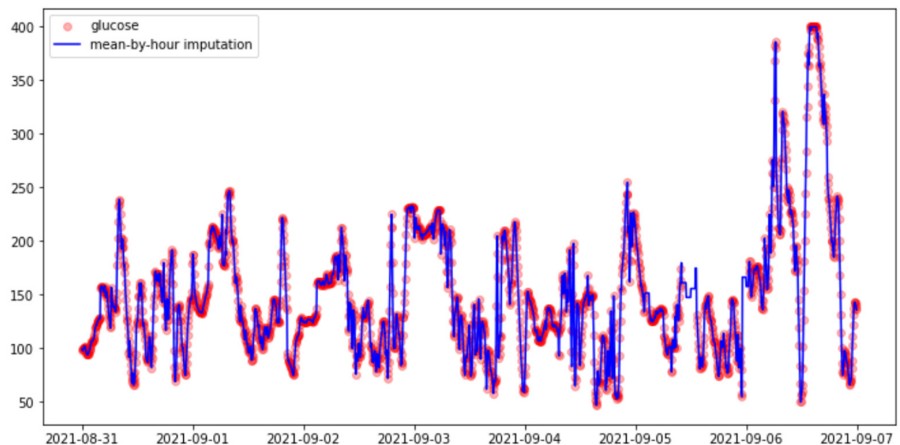

**Figure 5.** Comparing the true CGM time series with the imputed CGM using hourly mean, in one week period for patient 588.

(ii)   Linear Interpolation.

A gap of missing values is filled out for a straight line joining the extremes of the gap. Thus, the missing value in a given timestamp is taken from this straight line. Figure 6 shows the observed and imputed values by linear interpolation for subject 588 in the week starting on 31 August and ending on 7 September. As in Figure 6, the imputed values through linear interpolation follow the time series trend.

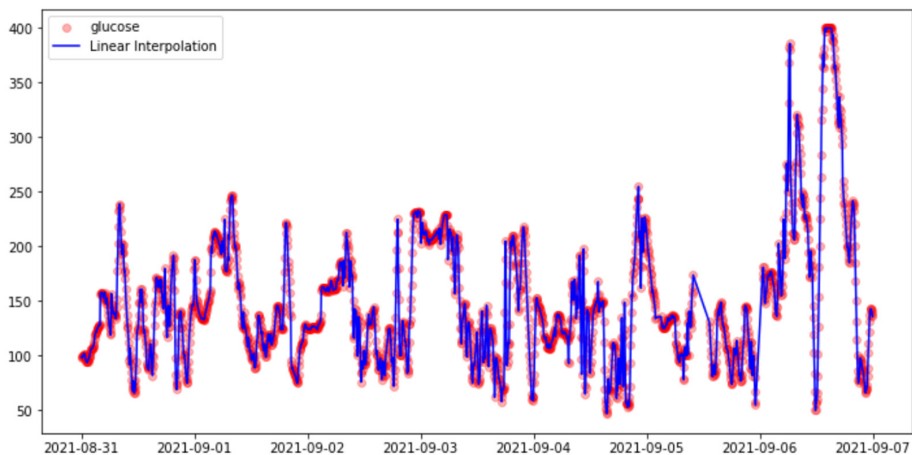

**Figure 6.** Comparing the true CGM time series with the imputed CGM using linear interpolation, in one week period for patient 588.

(iii) Spline Interpolation:

In this case, instead of simultaneously fitting a single, high-degree polynomial to all the values, spline interpolation fits low-degree polynomials to small subsets of the values. For instance, it fits five cubic polynomials between each pair of six points instead of fitting a single sixth-degree polynomial to all of them.

Cubic spline interpolation is the process of constructing a spline $f : [x_1, x_{n+1}] \rightarrow R$ that consists of $n$ polynomials of degree three, referred to as $f_1$ to $f_n$. A spline is a function defined by piecewise polynomials. Unlike regression, the interpolation function traverses all $n + 1$ pre-defined points of a dataset D. The resulting function has the following structure:

$$f(x) = \begin{cases} a_1x^3 + b_1x^2 + c_1x + d_1 & if\ x \in [x_1,\ x_2] \\ a_2x^3 + b_2x^2 + c_2x + d_2 & if\ x \in [x_2,\ x_3] \\ \quad\quad \dots \\ a_nx^3 + b_nx^2 + c_nx + d_n & if\ x \in [x_n,\ x_{n+1}] \end{cases} \tag{1}$$

Note that all polynomials are valid within an interval; they compose the interpolation function. Spline interpolation works only within the data boundaries $[x_1, x_{n+1}]$. With adequately chosen coefficients $a_i$, $b_i$, $c_i$ and $d_i$ for the polynomials, the resulting function traverses the points smoothly. Several equations are formulated to determine the coefficients, which all together compose a uniquely solvable system of equations. Figure 7 shows the observed and imputed values by spline interpolation for subject 588 in the week starting on 31 August and ending on 7 September. Figure 7 shows that imputed values by spline interpolation follow the trend less when we compare them with the two previous methods.

(iv) Polynomial Interpolation.

A gap of missing values is filled out by a polynomial of the lowest possible degree. In our study, we have used cubic polynomials.

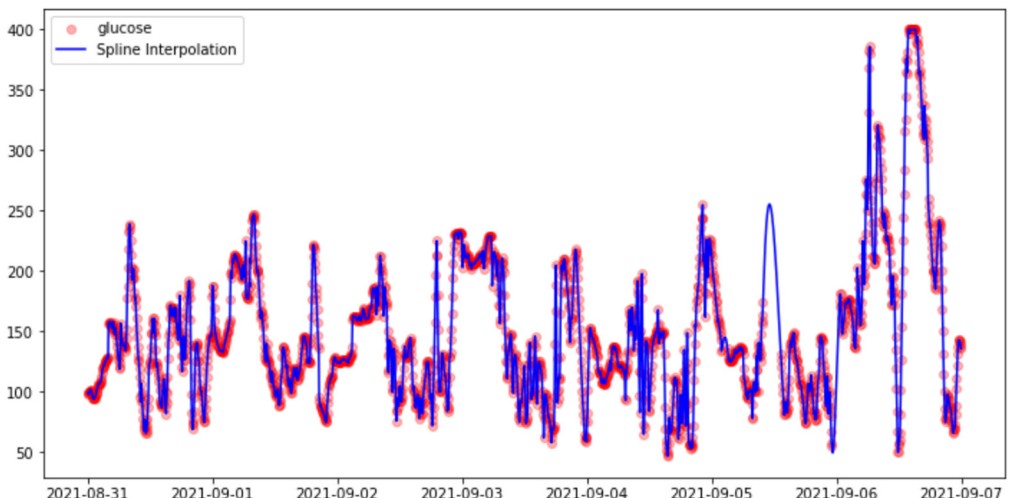

**Figure 7.** Comparing the true CGM time series with the imputed CGM using spline interpolation, in one-week period for patient 588.

### 3.3.3. Smoothing Methods

(i) Kalman Smoothing

The Kalman smoothing (KS) method outputs an interpolated time series of glucose estimates with mean and variance. It can automatically correct errors in the CGM readings where the estimated variance can be utilized for determining at which times the data are reliable. In our study, KS has been used as a pre-processing technique for sensor fault correction in the CGM reading. We use a modified implementation of KS for the OhioT1DM dataset, from the work [21,22]. The Kalman filter is a technique for estimating the current

state of a dynamical system from previous observations [23,24]. In Kalman filtering, records of data are used for the calculation of the estimates. Thus, the Kalman filter is appropriate for real-time data processing. It is a forward algorithm where each step is computed analytically. The model and observation can be written as follows:

$$\text{State system}: x_{t+1} = Ax_t + w_t, \text{ where } w_t \sim N(0,Q) \text{ for } t = 1, \ldots T-1,$$
$$\text{Output measurements}: y_t = Hx_t + v_t, \text{ where } v_t \sim N(0,R) \text{ for } t = 1, \ldots T. \tag{2}$$

We assume that the process state; $x_t$, the process noise; $w_t$, the sensor measurement; $y_t$, and the sensor noise; $v_t$ are independent. Note that all random variables above are either Gaussians or linear transformations of Gaussians and are, therefore, all Gaussian. The symmetric positive definite matrices $Q$ and $R$ are the covariance matrices of the process and the sensor noise, respectively. $A$ is the transition matrix and $H$ is the measurement matrix.

The goal of smoothing is to reconstruct or approximate the missing measurements given the known ones. Since the outputs and states are jointly Gaussian, the maximum likelihood and conditional mean estimates of the missing output values are the same. They can be found as the solution to the constrained least squares problem.

$$\text{Minimize } \sum_{i=1}^{T-1} \left\| Q^{-\frac{1}{2}}(\hat{x}_{t+1} - A\hat{x}_t) \right\|_2^2 + \sum_{i=1}^{T} \left\| R^{-\frac{1}{2}}(\hat{y}_t - H\hat{x}_t) \right\|_2^2$$
$$\text{subject to } (\hat{y}_t)_i = (y_t)_i, \ (t,i) \in K \tag{3}$$

where $K$ is the set of available outputs. For $(t,i) \notin K$, we take $(y_t)_i = NA$. We refer to entries of $y_t$ that are real as known measurements and entries of $y_t$ that have the value NA as missing measurements. Figure 8 shows the observed and smoothed values by Kalman smoothing for subject 588 in the week starting on 31 August and ending on 7 September. The Figure 8 suggests that the smoothing has generated some bias.

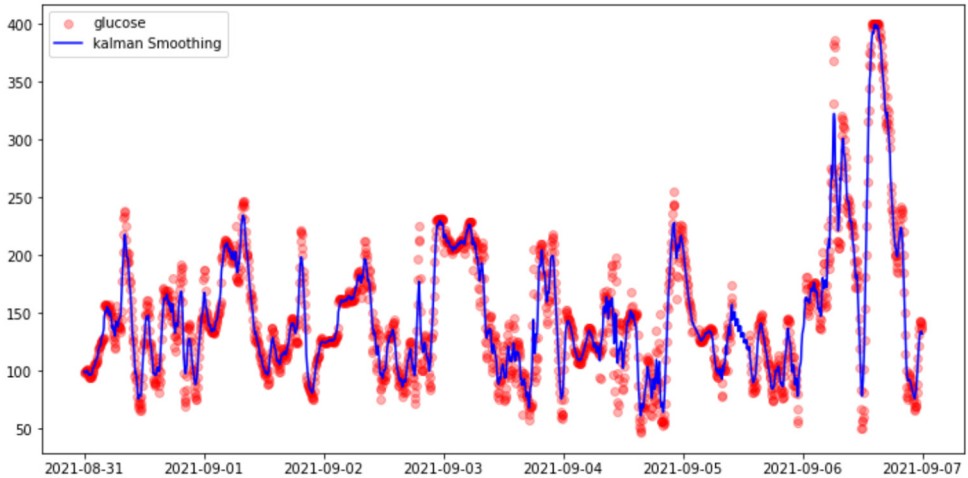

**Figure 8.** Comparing the true CGM time series with the smoothed CGM time series using Kalman smoothing, in one week period for patient 588.

(ii)   Smoothing Splines.

Cubic splines are piecewise cubic functions that are continuous and have continuous first and second derivatives. The continuity in all their lower-order derivatives makes splines very smooth.

The most natural way to parametrize the set of splines with knots at a given set of points $t_1, \ldots, t_m$ is to use the truncated power basis, $g_1, \ldots, g_{m+k+1}$, defined as $g_1(x) = 1$, $g_2(x) = x, \ldots, g_{k+1}(x) = x^k$, $g_{k+1+j}(x) = (x - t_j)_+^k$, $j = 1, \ldots m$. Here, $x_+$ denotes the positive part of $x$, i.e., $x_+ = \max\{x,0\}$. While these basis functions are natural, a

much better computational choice, both for speed and numerical accuracy, is the B-spline basis [25].

Regression splines are defined as $\hat{r}(x) = \sum_{j=1}^{m+k} \beta_j g_j(x)$ when $m = 4$, cubic splines are obtained, $k$ denotes the number of knots and $g_1, \ldots g_n$ are the truncated power basis functions for natural cubic splines with knots at $t_1, \ldots t_m$. The $\beta_j$ coefficients are obtained by least squares. That is, by minimizing,

$$\sum_{i=1}^{n}(y_i - \sum_{j=1}^{m}\beta_j g_j(x_i))^2 \tag{4}$$

Smoothing splines are defined as $\hat{r}(x) = \sum_{j=1}^{m} \beta_j g_j(x)$ where the vector $\beta$ of coefficients is obtained by minimizing $||y - G\beta||_2^2 + \lambda\beta^T\acute{\Omega}\beta$, $g_1, \ldots g_n$ are the truncated power basis functions for natural cubic splines with knots at $x_1, \ldots x_n$. Here $G$ is the basis matrix and $\acute{\Omega}$ is the penalty matrix defined as

$$\Omega_{ij} = \int g_i''(t)g_j''(t)dt \text{ for } i,j = 1,2,\ldots,n \tag{5}$$

The smoothing splines can also be obtained by minimizing:

$$\sum_{i=1}^{n}(y_i - f(x_i)^2 + \lambda \int (f''(x)^2 dx \tag{6}$$

$\lambda \geq 0$ is a smoothing parameter, when $\lambda \to 0$ the smoothing spline converges to the interpolation spline, when $\lambda \to \infty$ the smoothing splines converges to linear least squares estimate.

In this paper, we have used the smoothing parameter spar, such that $\lambda = r256^{3spar-1}$ where $r = tr(X'WX)/tr(\Sigma)$, $\Sigma$ is the penalty matrix associated with a B-splines basis, $X$ is given by $X_{i,j} = B_j(x_i)$, $W$ is the diagonal matrix of weights (scaled such that its trace is $n$, the original number of observations), and $B_k(.)$, is the *k-th* B-spline. Figure 9 shows the observed and the smoothed values by smoothing spline for the subject 588 in the week starting on 31 August and ending 7 September. Clearly, the smoothing spline has generated bias.

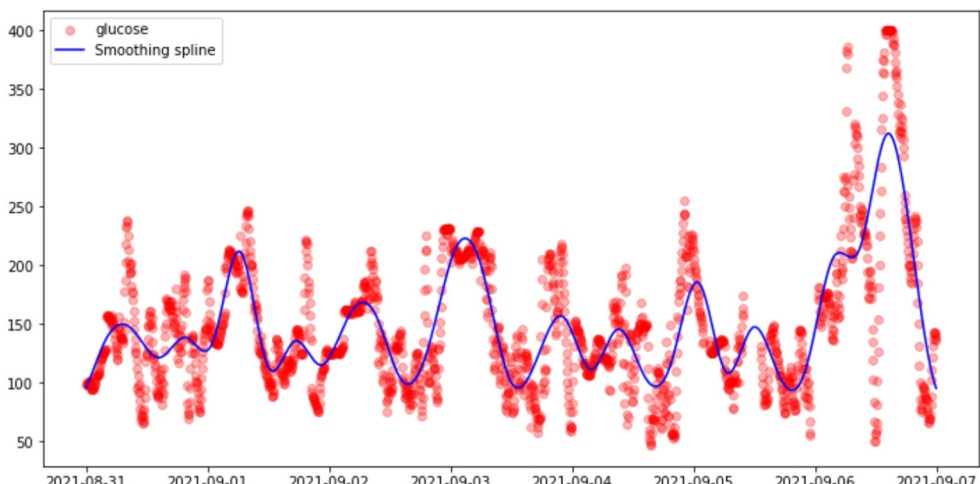

**Figure 9.** Comparing the true CGM time series with the smoothed CGM time series using smoothing spline, in one week period for patient 588.

*3.4. Prediction Models*

3.4.1. XGBoosting

XGBoosting (short for Extreme Gradient Boosting) is an efficient implementation of gradient boosting for classification and regression problems. It is both fast and efficient, performing well. It is an ensemble of decision tree algorithm where new trees fix errors of

those trees already part of the model. Trees are added until no further improvements can be made to the model [26].

XGBoosting can also be used for time series forecasting, although it requires that the time series dataset be transformed into a supervised learning problem first. Midroni et al. [7] and Allfian et al. [27] applied XGBoosting to predict glucose levels in five patients using a prediction horizon of 30 and 60 min. XGBoosting does not support multi-target regression, so we separately predict the glucose level for each of the six future timesteps and then ensemble them in one prediction matrix.

### 3.4.2. One-Dimensional Convolutional Neural Networks (1D-CNN)

A convolution can be thought of a 'weighted sum of memories'. Suppose that $f(t)$ represents sound at time $t$ and $h(\tau)$ is the proportion one heard from $\tau$ seconds ago, and that one can only hear sound at discrete time steps. Then what you hear at time can be represented as:

$$h(0)f(t) + h(1)f(t-1) + h(2)f(t-2) + \cdots + h(t)f(0) \tag{7}$$

Note that this is a weighted moving average, where the weights are given by the function. Thus, a discrete-time convolution generalizes a moving average so that the weights are non-zero and may not sum to 1. Like a moving average, a convolution smooths a time series.

The basic architecture of CNN cannot be applied for usual time series data prediction since the CNN structures are 2D-CNN, which only take 2D inputs. Therefore, the conventional 2D-CNN architecture is not directly applicable to 1D signal prediction. Some researchers have converted 1D signals into 2D images to use the 2D-CNN architectures directly. However, in most common cases, it increases computational costs and decreases efficiency. The significant advantages of 1D-CNN are that it requires much less computational complexity and time than 2D-CNN and takes 1D signal directly without 2D conversion. Due to these advantages, there are many studies and applications of 1D-CNN. Using 1D-CNN, correlational properties of multivariate signals can be extracted without additional feature engineering. Bhimireddy et al. [8] used a hybrid 1D-CNN and LSTM model to predict glucose levels on the OhioT1DM dataset. A bi-directional LSTM model outperformed the proposed model. They only got an RMSE of 20.6 and did not consider data imputation to handle missing values. Figure 10 shows the architecture of the 1D-CNN model used in this paper.

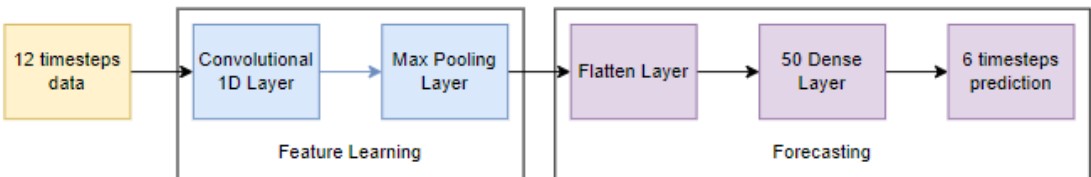

**Figure 10.** The 1D-CNN architecture used to predict blood glucose levels.

### 3.4.3. Transformers

Vaswani et al. [28] outlined the concept of attention-based networks, originally in the context of natural language processing. NLP deals with sequences of words ordered by grammar and syntax. The attention-based network, also known as Transformer, takes an input text sequence, for example, in English, and generates an output text sequence, for instance, in Spanish. Time series analysis works with chronologically ordered sequences: time steps. The Transformer must generate a forecast sequence along the time axis from a sequence of training observations. Transformers capture long-range dependencies and interactions, making them attractive for time series modeling. In several applications, Transformers outperformed RNN and LSTM models [6].

The attention heads enable the Transformer to learn relationships between a time step and every other time step in the input sequence. The Transformer updates its attention weights and downgrades the least relevant time steps. A score matrix expresses how closely other time steps are associated with the time step in question.

At its core, it contains a stack of Encoder layers and Decoder layers. The Encoder stack and the Decoder stack each have corresponding Embedding layers for their respective inputs. Finally, there is an Output layer to generate the final output.

Each Attention Block consist of Self Attention, Layer Normalization, and a Feed-Forward layer. The input dimensions of each block are equal to its output dimensions.

## 4. Results and Discussion

We compared three algorithms in three situations: in the first one, only CGM is used to predict glucose level without carrying out imputation. Thus, instances with at least one missing value are deleted. This approach is used in most of the studies about blood glucose prediction. In the second situation, only CGM is used to predict glucose level, and either imputation or smoothing is used. In the third case, two features: CGM and meals are used to predict glucose level, and either imputation or smoothing is used.

The metric used to evaluate the performance of our algorithms was the root mean squared error (*RMSE*) defined by:

$$RMSE = \sqrt{\frac{\sum_{i=1}^{n}(y - \hat{y})^2}{n}} \tag{8}$$

There are other metrics, such as MAE and the symmetric mean absolute percentage error (SMAPE), but *RMSE* is the most used.

### 4.1. Using Only CGM without Imputation to Predict Glucose Level

After normalizing the data using Max-Min scaling, where the data are mapped to the interval [0, 1], each of the three algorithms is repeated five times, and the mean and the standard deviation of the five RMSEs were obtained. In the tables below, we have included only the RMSE average. All our experiments were implemented in Python 3.8 along with scikit-learn, keras, and Tensorflow 2.0.

The XGBoosting model was run using up to 5000 iterations. The 1D-CNN algorithm was run according to the architecture shown in Figure 10, with 100 epochs considering a batch size of 32 and 20 percent of the validation set. The Adam (adaptive moment estimation) optimizer was used in the compilation process. The Transformer algorithm was run with 50 epochs following the architecture shown in Figure 11. The batch size was 64, the number of transformer blocks was 4 with 4 heads, each of them of size 256, and the Adam optimizer was used to compile the model.

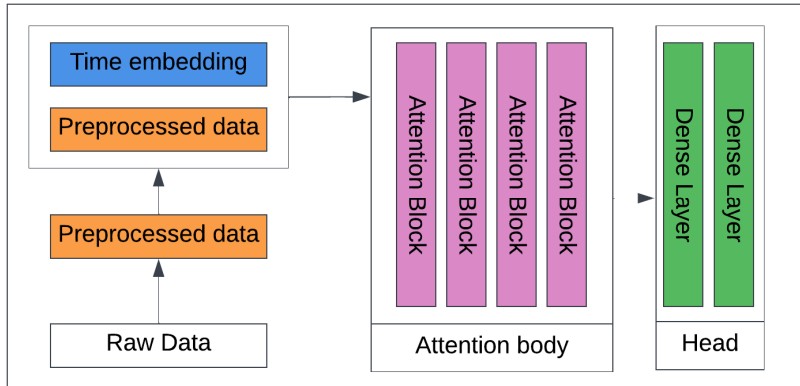

**Figure 11.** The Transformer architecture used to predict blood glucose level.

As shown in Table 3, the Transformer prediction algorithm has the best performance since it has the lowest RMSE average. Besides that, it offers low variability and fast computation time. Averaging the RMSE of the three algorithms in each dataset, we found that subject 570 has the lowest RMSE, whereas subject 575 has the highest RMSE.

**Table 3.** RMSE for prediction models using only CGM as predictor without carrying out imputation.

| Subject | XGBoosting | 1D-CNN | Transformer |
|---------|------------|--------|-------------|
| 559 | 14.14 | 15.96 | 13.57 |
| 563 | 14.10 | 16.63 | 13.40 |
| 570 | 11.95 | 13.19 | 11.34 |
| 575 | 16.80 | 19.13 | 16.78 |
| 588 | 13.53 | 15.72 | 12.89 |
| 591 | 15.22 | 17.87 | 14.32 |
| 540 | 16.25 | 17.15 | 14.02 |
| 584 | 16.16 | 18.28 | 15.21 |
| **Average** | **14.77** | **16.74** | **13.94** |

Next, we will compare the performance of the two smoothing methods and the four imputation methods for each of the three algorithms.

According to Table 4 for the XGBoosting, on average, the best of the two smoothing methods was Kalman smoothing, whereas, among the imputation methods, all the interpolation methods gave a similar average of RMSE. On average, the hourly mean gave the highest RMSE. Taking the average of the four imputation methods, patient 540 has the highest RMSE value, whereas patient 570 has the lowest RMSE among all the datasets. Additionally, the hourly mean imputation does not perform well for subject 540. The reason could be that there is no effect of the hour of the day on the glucose levels for this subject. In all the imputation methods, except the hourly mean imputation, subject 570 has the lowest RMSE. When hourly mean imputation is used, subject 588 has the lowest RMSE, followed by subject 563 and subject 570.

**Table 4.** RMSE for predicting glucose level using univariate XGBoosting after imputation/smoothing.

| Subject | Smoothing | | Imputation | | | |
|---------|-----------|-----------|------------|-----------|------------|----------------|
| | | | Interpolation | | | |
| | Kalman | Spline | Linear | Spline | Polynomial | Hourly Mean |
| 559 | 7.07 | 11.55 | 13.16 | 13.36 | 13.60 | 16.44 |
| 563 | 6.48 | 6.38 | 13.78 | 13.71 | 13.73 | 14.17 |
| 570 | 6.39 | 7.59 | 11.91 | 11.74 | 11.51 | 16.15 |
| 575 | 8.25 | 8.45 | 16.23 | 16.20 | 16.32 | 20.38 |
| 588 | 7.00 | 7.08 | 13.46 | 13.53 | 13.68 | 13.75 |
| 591 | 7.35 | 7.46 | 15.26 | 15.34 | 15.43 | 16.56 |
| 540 | 10.13 | 16.07 | 17.87 | 17.25 | 17.57 | 31.30 |
| 584 | 7.47 | 8.97 | 15.55 | 15.46 | 15.42 | 20.23 |
| **Average** | **7.51** | **9.19** | **14.43** | **14.57** | **14.65** | **18.62** |

The computation of the RMSE for the XGBoosting model prediction is fast.

The 1D-CNN algorithm was run with 200 epochs considering a batch size of 32 and 20 percent of validation set = 0.2.

Table 5 shows that when CNN-1D is used, the smoothing spline gives the best results among the two smoothing methods. Among the imputation methods, linear interpolation has the lowest RMSE average. As in XGboosting, subject 540 has the highest RMSE value among all the subjects, whereas subject 570 has the lowest RMSE. Similarly to the previous case, in all the imputation methods, except the hourly mean imputation, patient 570 has the lowest RMSE. When hourly mean imputation is used, subject 570 has the third lowest RMSE. Subject 588 has the lowest RMSE, followed by subject 563.

**Table 5.** Average RMSE for predicting glucose level using 1D-CNN after imputation/smoothing.

| Subject | Smoothing | | Imputation | | | |
| | | | Interpolation | | | |
| | Kalman | Spline | Linear | Spline | Polynomial | Hourly Mean |
|---|---|---|---|---|---|---|
| 559 | 7.94 | 5.51 | 14.81 | 16.25 | 16.40 | 17.49 |
| 563 | 8.41 | 5.12 | 16.36 | 17.07 | 18.12 | 16.15 |
| 570 | 6.66 | 4.63 | 13.41 | 13.61 | 12.90 | 17.04 |
| 575 | 9.15 | 6.53 | 19.05 | 19.18 | 18.27 | 21.74 |
| 588 | 8.27 | 5.44 | 13.58 | 18.35 | 18.70 | 15.48 |
| 591 | 8.98 | 6.33 | 17.90 | 20.22 | 19.52 | 18.75 |
| 540 | 8.57 | 7.24 | 16.23 | 20.30 | 19.73 | 27.40 |
| 584 | 8.82 | 8.23 | 17.53 | 21.00 | 21.31 | 22.07 |
| **Average** | 8.35 | 6.12 | 16.10 | 18.24 | 18.11 | 19.51 |

Table 6 shows that for the Transformer algorithm, Kalman smoothing gives better results than smoothing spline. Among the imputation methods, linear interpolation gives the best RMSE. All the other three methods perform almost the same. On average, the highest RMSE value was obtained for subject 584, whereas the lowest RMSE was obtained for subject 570.

**Table 6.** Average RMSE for predicting glucose level using Transformer after imputation/smoothing.

| Subject | Smoothing | | Imputation | | | |
| | | | Interpolation | | | |
| | Kalman | Spline | Linear | Spline | Polynomial | Hourly Mean |
|---|---|---|---|---|---|---|
| 559 | 6.93 | 9.30 | 12.13 | 15.31 | 15.34 | 14.87 |
| 563 | 7.40 | 9.38 | 12.95 | 13.37 | 13.26 | 13.38 |
| 570 | 7.27 | 8.96 | 11.20 | 11.87 | 11.95 | 14.91 |
| 575 | 8.57 | 12.13 | 16.11 | 17.15 | 17.09 | 19.10 |
| 588 | 7.67 | 10.48 | 12.52 | 12.91 | 13.00 | 12.91 |
| 591 | 8.78 | 11.59 | 14.41 | 14.11 | 14.95 | 15.24 |
| 540 | 8.05 | 13.08 | 13.57 | 15.04 | 15.18 | 16.19 |
| 584 | 8.85 | 14.05 | 14.51 | 19.32 | 19.25 | 18.83 |
| **Average** | 7.94 | 11.12 | 13.42 | 14.88 | 15.00 | 15.67 |

Since stationarity can affect the performance of Transformers after imputation was performed [29], we computed the Augmented Dickey–Fuller test to check the stationarity of each time series, and we obtained test values more significant than the critical values, concluding that all the eight-time series were stationary.

*4.2. Using CGM and Meal to Predict Glucose Level*

In the XGBoosting algorithm, we used 24 features as predictors; 12 were from CGM at the 12 previous timesteps, and the remaining 12 were from the meal feature at the respective timesteps.

Comparing the results from Tables 4 and 7, we can note that when the meal feature is included, the RMSE for the XGBoosting algorithm decreases by a small amount only when imputation by hourly mean is used. It has a small increment when smoothing spline is used; in all the remaining methods, the RMSE remains almost the same. Similarly, when only CGM is used, the Kalman smoothing and the linear interpolation give, on average, the lowest RMSE. Additionally, subject 570 has the lowest RMSE average, whereas subject 540 has the highest RMSE average when imputation is applied.

**Table 7.** Average RMSE for predicting glucose level using XGBoosting after imputation/smoothing.

| Subject | Smoothing | | Imputation | | | |
| --- | --- | --- | --- | --- | --- | --- |
| | | | Interpolation | | | |
| | Kalman | Spline | Linear | Spline | Polynomial | Hourly Mean |
| 559 | 7.20 | 11.73 | 13.65 | 12.99 | 13.32 | 16.69 |
| 563 | 6.62 | 6.62 | 13.86 | 13.81 | 13.81 | 14.21 |
| 570 | 6.43 | 7.90 | 11.70 | 11.37 | 11.26 | 15.89 |
| 575 | 8.49 | 8.92 | 16.69 | 16.58 | 16.57 | 20.68 |
| 588 | 7.21 | 7.58 | 13.53 | 13.37 | 13.56 | 13.72 |
| 591 | 7.34 | 7.84 | 14.98 | 15.14 | 15.32 | 16.31 |
| 540 | 10.41 | 18.03 | 16.37 | 17.77 | 18.04 | 31.58 |
| 584 | 7.64 | 9.08 | 15.53 | 15.47 | 15.49 | 20.15 |
| **Average** | 7.66 | 9.71 | 14.53 | 14.56 | 14.67 | 18.65 |

A 1D-CNN layer followed by a Max-Pooling layer is applied to each feature. Then, the result is concatenated, and a dense layer with 100 neurons is applied. After that, the forecasting for the future 30 min (six timesteps) is carried out.

From Tables 5 and 8, when meal is included, the RMSE for the 1D-CNN decreases only a small amount when smoothing spline is used. In all the remaining methods, the RMSE increases and the most significant increment occurs when spline interpolation is used. Smoothing spline gives lower RMSE than Kalman smoothing. Averaging the four imputation methods, we obtained the lowest RMSE for dataset 570, whereas subject 584 had the highest RMSE.

**Table 8.** Average RMSE for predicting glucose level using bivariate (CGM and meal) 1D-CNN after imputation/smoothing.

| Subject | Smoothing | | Imputation | | | |
| --- | --- | --- | --- | --- | --- | --- |
| | | | Interpolation | | | |
| | Kalman | Spline | Linear | Spline | Polynomial | Hourly Mean |
| 559 | 8.53 | 5.55 | 21.12 | 24.91 | 21.86 | 25.08 |
| 563 | 8.23 | 4.76 | 20.84 | 21.19 | 18.35 | 18.60 |
| 570 | 7.18 | 4.56 | 15.47 | 15.54 | 14.72 | 18.77 |
| 575 | 9.34 | 6.89 | 22.57 | 24.43 | 21.02 | 26.16 |
| 588 | 11.03 | 5.02 | 21.14 | 22.50 | 23.00 | 21.10 |
| 591 | 9.50 | 6.16 | 21.63 | 23.06 | 21.47 | 21.62 |
| 540 | 8.64 | 7.36 | 18.26 | 24.47 | 21.88 | 27.40 |
| 584 | 8.87 | 7.31 | 18.58 | 30.55 | 24.80 | 24.08 |
| **Average** | 8.91 | 5.95 | 19.95 | 23.33 | 20.88 | 22.85 |

Including meals as a predictor does not improve the prediction model's performance.

Before using meals with CGM to predict glucose level with the Transformer model, we removed any non-stationarity effect that the time series for meals could generate. This effect is achieved by differentiating the meal more than one time. It was necessary to differentiate the time series for meals two times.

Tables 6 and 9 show that when meals are included, the RMSE for the Transformers decreases greatly when smoothing spline is used. In all the remaining methods, the RMSE increases, and the most significant increment occurs when linear interpolation is used. Smoothing spline gives lower RMSE than Kalman smoothing. Averaging the four imputation methods in each dataset, we obtained the lowest RMSE average for dataset 563, whereas subject 575 provided the highest RMSE average.

As in 1D-CNN, the inclusion of meal as a predictor does not improve the performance of the Transformer algorithm.

**Table 9.** Average RMSE for predicting glucose level using bivariate (CGM and meal) Transformer after imputation smoothing.

| Subject | Smoothing | | Imputation | | | |
| | | | Interpolation | | | |
| | Kalman | Spline | Linear | Spline | Polynomial | Hourly Mean |
|---|---|---|---|---|---|---|
| 559 | 9.49 | 5.40 | 13.99 | 14.60 | 14.48 | 15.33 |
| 563 | 8.97 | 4.99 | 13.83 | 13.65 | 13.49 | 14.11 |
| 570 | 10.60 | 6.98 | 13.14 | 13.71 | 13.41 | 17.23 |
| 575 | 11.73 | 6.63 | 18.52 | 18.85 | 18.49 | 21.87 |
| 588 | 9.98 | 6.31 | 14.60 | 15.30 | 15.38 | 15.02 |
| 591 | 10.87 | 5.52 | 16.48 | 17.19 | 16.96 | 17.20 |
| 540 | 12.00 | 6.66 | 16.45 | 16.05 | 16.13 | 18.11 |
| 584 | 11.30 | 5.31 | 16.67 | 16.57 | 16.59 | 20.00 |
| **Average** | 10.61 | 5.97 | 15.46 | 15.74 | 15.61 | 17.35 |

## 5. Conclusions and Future Work

After running our experiments, we reached the following conclusions:

First, the Transformer algorithm outperforms XGBoosting and 1D-CNN algorithms when only CGM is used to predict blood glucose level and neither imputation nor smoothing is applied to the data.

Second, when only CGM is used as a predictor, linear interpolation appears to be the best imputation method. This fact is clearly observed in the deep learning algorithms, yet it is not as evident in the XGBoost model.

Third, when imputation is applied and only CGM is used as a predictor, the Transformer model and the XGBoosting model perform almost the same, whereas the 1D-CNN model does not perform well.

Fourth, using only CGM as a predictor, Kalman smoothing yields better results than smoothing splines for the Transformer and XGBoost algorithms, but smoothing splines perform better for 1D-CNN.

Fifth, applying imputation, including a second feature (meal), increased the RMSE of the deep learning prediction models. Only the XGBoost algorithm was not affected. Midroni et al. also noticed this last result [7].

Sixth, when smoothing is applied, including a second feature does not affect the RMSE of either the smoothing method for XGBoost or 1D-CNN. However, Transformer's RMSE is significantly reduced when smoothing splines are applied, while its RMSE increases with Kalman smoothing.

In general, our results outperform the ones mentioned in the related work section of this paper. When smoothing is applied to the data, our results using deep learning models are better than the one mentioned in Rabby et al. [16].

Finally, the generated bias can affect the performance of the models on the considered subjects.

One limitation in our work has been the lack of information on some interesting features in the second cohort of the OhioT1DM Dataset. Because of that, we did not include additional features in our study.

In future work, we plan to investigate the effect of bias on these prediction models' performance. Additionally, we would like to understand the stationarity's impact on Transformer further. Lastly, we would like to explore more complex architecture for deep learning models to predict blood glucose levels.

**Author Contributions:** Conceptualization, E.A.; methodology, E.A.; software, E.A. and R.A.; validation, E.A., R.A. and V.P.; investigation, E.A. and R.A.; data curation, E.A. and R.A.; writing—original draft preparation, E.A.; writing—review and editing, E.A., R.A. and V.P.; visualization, E.A. and R.A. All authors have read and agreed to the published version of the manuscript.

**Funding:** This research received no external funding.

**Institutional Review Board Statement:** Not applicable.

**Informed Consent Statement:** Not applicable.

**Data Availability Statement:** The original data OhioT1DM can be obtained from the University of Ohio upon request. The preprocessed data used in this paper are available at github.com/eacunafer/glucose-prediction accessed on accessed on 13 July 2021.

**Conflicts of Interest:** The authors declare no conflict of interest.

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
