# Peer review of "Analyzing the Performance of Transformers for the Prediction of the Blood Glucose Level Considering Imputation and Smoothing"

_2504-2289, doi:10.3390/bdcc7010041_

Round 1
Reviewer 1 Report
The article presents a work that aims to investigate the effect of pre-processing techniques, data imputation and smoothing, in the future prediction (30 min horizon) of the blood glucose level in patients with level 1 diabetes, using Transformers. Tests were performed using the Dataset: OhioT1DM (History containing time series lasting 60 min). A comparison was performed using 3 models for the task: Transformers, CNN 1D, XGBoost.
Experiments: Three trainings were performed, the first using only CGM without “imputation”, the second using only CGM data with “imputation/smoothing” and the third using CGM and meal data with “imputation/smoothing”. The RMSE metric was chosen for the tests.
Points for improvement:
a) Include comparison of methods and results with related work;
b) Results tables a bit confusing (4 to 9), perhaps you could consider separating the Smoothing results from the Interpolation results.
c) There could be the use of some optimization technique to define hyper parameters of CNN and the Transformers Network.
d) The lowest RMSE means were obtained using CNN with the spline smoothing technique, however, this does not seem to have been much explored by the authors.
e) Improve the discussion of the results as a whole, adding comparative analyzes and case studies.
f) Clearly point out the contribution of the work in the introduction and conclusion, address the extent to which the intended contribution has been achieved
Author Response
- We made improvements to the results and discussion section
- We clarify the presentations of tables 4-9.
- We include the name of the optimizer used to compiler 1D-CNN and Transformers algorithms.
- 1D-CNN outperforms XGBoosting and Transformers only in one of the six main comparison that we made and that it is mentioned in the conclusion section.
- The Results and discussion section was improved.
- We split the introduction section in two new sections: Introduction and Related work, and in the introduction section we point out the contribution of our work. In the conclusion section we mention our achievements.
Reviewer 2 Report
#Write motivation, novelty and contribution of the study in the introduction section.
#Write a separate related work section. You may read and cite these books: Healthcare data analytics and management, Medical Big Data and internet of medical things: Advances, challenges and applications etc.. Highlight the research gap. Add a section organization at the end of the related work section.
# [x1,xn+1], ai, bi, ci, and di having issue of sub-script. All eqn should be written in italics. Give eqn number(s).
#Write the motivation of choosing 1D-CNN. Why 1D-CNN out performing others.
#Table 3 - correct sequence will be Subject, XGBoosting, Transformer , CNN-1D
#Add a detailed discussion section.
#Write limitations of the study in conclusion section. Write conclusion in a single para.
Author Response
- We split the introduction section in two new sections: Introduction and Related work, and in the introduction section we point out the motivation, novelty, contribution of our work.
- We have included a related work section
- We corrected the subscripts notation an add number to all the equations appearing in the paper.
- The motivation to use 1D-CNN appears in the description of the algorithm, see subsection 3.4.2
- We think that the ordering of the header in Table 3 is correct.
- There is a Results and discussion section, which was improved.
- Limitations of the study was included in the conclusion section